# Efficacy and Safety of Bevacizumab Plus Erlotinib in Patients with Renal Medullary Carcinoma

**DOI:** 10.3390/cancers13092170

**Published:** 2021-04-30

**Authors:** Andrew J. Wiele, Devaki Shilpa Surasi, Priya Rao, Kanishka Sircar, Xiaoping Su, Tharakeswara K. Bathala, Amishi Y. Shah, Eric Jonasch, Vince D. Cataldo, Giannicola Genovese, Jose A. Karam, Christopher G. Wood, Nizar M. Tannir, Pavlos Msaouel

**Affiliations:** 1Division of Cancer Medicine, The University of Texas MD Anderson Cancer Center, Houston, TX 77030, USA; AJWiele@mdanderson.org; 2Department of Nuclear Imaging, Division of Diagnostic Imaging, The University of Texas MD Anderson Cancer Center, Houston, TX 77030, USA; DSSurasi@mdanderson.org; 3Department of Pathology, Division of Pathology and Laboratory Medicine, The University of Texas MD Anderson Cancer Center, Houston, TX 77030, USA; PRao@mdanderson.org (P.R.); KSircar@mdanderson.org (K.S.); 4Department of Bioinformatics and Computational Biology, Division of Quantitative Sciences, The University of Texas MD Anderson Cancer Center, Houston, TX 77030, USA; XSu1@mdanderson.org; 5Department of Abdominal Imaging, Division of Diagnostic Imaging, The University of Texas MD Anderson Cancer Center, Houston, TX 77030, USA; TKBathala@mdanderson.org; 6Department of Genitourinary Medical Oncology, The University of Texas MD Anderson Cancer Center, Houston, TX 77030, USA; AYShah@mdanderson.org (A.Y.S.); EJonasch@mdanderson.org (E.J.); GGenovese@mdanderson.org (G.G.); 7Mary Bird Perkins—Our Lady of the Lake Cancer Center, Baton Rouge, LA 70809, USA; vince.cataldo@fmolhs.org; 8Department of Genomic Medicine, Division of Cancer Medicine, The University of Texas MD Anderson Cancer Center, Houston, TX 77030, USA; 9Department of Urology, Division of Surgery, The University of Texas MD Anderson Cancer Center, Houston, TX 77030, USA; JAKaram@mdanderson.org (J.A.K.); CGWood@mdanderson.org (C.G.W.); 10Department of Translational Molecular Pathology, Division of Pathology and Laboratory Medicine, The University of Texas MD Anderson Cancer Center, Houston, TX 77030, USA

**Keywords:** renal medullary carcinoma, bevacizumab, erlotinib, platinum-based chemotherapy, aerobic glycolysis

## Abstract

**Simple Summary:**

Renal medullary carcinoma (RMC) is a rare and highly aggressive renal cell carcinoma, with a median survival of 13 months. Platinum-based chemotherapy is the recommended standard of care for RMC, but no effective salvage regimens have been established to date. Previous comprehensive molecular characterization of RMC tissues revealed a reliance on aerobic glycolysis, suggesting that bevacizumab plus erlotinib may be an effective regimen against RMC. The aim of our retrospective study was to evaluate the efficacy and safety of bevacizumab plus erlotinib in patients with RMC. In ten patients, the combination was safe and effective, establishing bevacizumab plus erlotinib as a new salvage regimen in RMC.

**Abstract:**

Purpose: To assess the efficacy and safety of bevacizumab plus erlotinib in patients with RMC. Methods: We retrospectively reviewed the records of patients with RMC treated with bevacizumab plus erlotinib at our institution. Results: Ten patients were included in the study. Two patients achieved a partial response (20%) and seven patients achieved stable disease (70%). Tumor burden was reduced in seven patients (70%) in total, and in three out of five patients (60%) that had received three or more prior therapies. The median progression-free survival (PFS) was 3.5 months (95% CI, 1.8–5.2). The median overall survival (OS) from bevacizumab plus erlotinib initiation was 7.3 months (95% CI, 0.73–13.8) and the median OS from diagnosis was 20.8 months (95% CI, 14.7–26.8). Bevacizumab plus erlotinib was well tolerated with no grade ≥4 adverse events and one grade 3 skin rash. Dose reduction was required in one patient (10%). Conclusions: Bevacizumab plus erlotinib is clinically active and well tolerated in heavily pre-treated patients with RMC and should be considered a viable salvage strategy for this lethal disease.

## 1. Introduction

Renal medullary carcinoma (RMC) is a rare and highly aggressive malignancy that is uniformly associated with sickle hemoglobinopathies and most frequently affects young males of African descent [1,2]. The median survival is 13 months, less than 5% of RMC patients survive longer than 36 months, and cytotoxic chemotherapy produces typically brief objective responses in only 29% of patients [2,3]. Platinum-based chemotherapy is currently the recommended standard of care, and the combination of gemcitabine with doxorubicin has also shown clinical activity in RMC [2,3]. Furthermore, studies have suggested that RMC is refractory to the anti-angiogenic tyrosine kinase inhibitor therapies used in patients with clear cell renal cell carcinoma [2,4,5].

RMC is characterized by the loss of SMARCB1 (INI-1) nuclear staining, a potent tumor suppressor gene on chromosome 22 and critical subunit of the SWItch/Sucrose Non-Fermentable (SWI/SNF) complex [6,7]. The SWI/SNF complex hydrolyzes adenosine triphosphate to remodel chromatin structures, and SMARCB1 inactivation deregulates the SWI/SNF complex, resulting in aggressive tumorigenesis [6]. Comprehensive molecular characterization of RMC tissues revealed a decrease in genes related to the tricarboxylic acid (TCA) cycle and an increase in pyruvate dehydrogenase kinase isozyme 1 (PDK1), which decreases the use of pyruvate in the TCA cycle, suggesting a reliance on aerobic glycolysis to meet cellular bioenergetic needs (Warburg Effect) [7,8]. RMC tissues also demonstrated increased expression of oncogenic pathways associated with aerobic glycolysis such as c-MYC signaling, the epidermal growth factor receptor (EGFR) pathway (Appendix A), and the glucose transporter 1 (GLUT1) gene [7,8,9]. Clinically, the increased utilization of glucose by RMC tumors is demonstrated by the high uptake of fluorodeoxyglucose (FDG) on positron emission tomography (PET) imaging (Appendix A), similarly to hereditary leiomyomatosis and renal cell cancer (HLRCC), which also relies heavily on aerobic glycolysis [5,8,10,11,12].

The combination of the vascular endothelial growth factor (VEGF) inhibitor bevacizumab with the EGFR inhibitor erlotinib impairs glucose uptake by cancer cells, and it has accordingly been shown to be particularly effective in HLRCC compared with tumors that are less dependent on aerobic glycolysis [8,10]. Additionally, this metabolic pathway is not directly related to cytotoxic chemotherapy resistance pathways [8,13]. We therefore hypothesized that bevacizumab plus erlotinib would show activity against RMC, including in patients with tumors resistant to prior cytotoxic chemotherapy. To test this hypothesis, we reviewed herein the clinical efficacy and safety of bevacizumab plus erlotinib in 10 patients with metastatic RMC.

## 2. Materials and Methods

### 2.1. Study Design

We conducted a single-institution retrospective review of patients with metastatic RMC that were treated with the combination of bevacizumab plus erlotinib from March 2005 through September 2020 at The University of Texas MD Anderson Cancer Center (MDACC). The study was approved by the MDACC Institutional Review Board (protocol PA16-0736). Baseline demographic and clinical data for each patient were collected by individual chart review from the institution’s electronic medical record system. All RMC samples were confirmed to be SMARCB1 negative by immunohistochemistry using purified mouse anti-BAF47 Clone 25/BAF47 (BD Biosciences, San Jose, CA, USA). Sickle cell status was determined by hemoglobin electrophoresis.

Patients were managed according to best practice at MDACC during bevacizumab plus erlotinib therapy. Charts were reviewed with attention to treatment dose adjustments, treatment discontinuations, and toxicities assessed according to the Common Terminology Criteria for Adverse Events (CTCAE v5). A board-certified radiologist (SDS), blinded to patient history and clinical data, assessed radiographic tumor response to bevacizumab plus erlotinib using Response Criteria in Solid Tumors (RECIST), version 1.1 [14]. Somatic alterations were identified in tumor tissues by PCR-amplicon-based target capture using Oncomine [15].

### 2.2. Statistical Analysis

Patient characteristics were summarized using median, interquartile range, and minimum/maximum values for continuous variables and frequency (%) for categorical variables. Progression free survival (PFS) was defined as the time interval between the date of first bevacizumab plus erlotinib dose and the date of disease progression or death from any cause, whichever occurred first. Patients who were alive without disease progression at the time of last follow-up were censored at the time of last follow-up. Overall survival 1 (OS1) was defined as the time interval between the date of first bevacizumab plus erlotinib dose and the date of death due to any cause. Overall survival 2 (OS2) was defined as the time interval between the date of RMC diagnosis and the date of death due to any cause. Patients that were alive were censored at the last follow-up. The Kaplan–Meier method was applied to estimate time-to-event outcomes.

## 3. Results

### 3.1. Demographic, Clinical, Molecular, and Treatment Characteristics

Between 03/2005 and 09/2020, 10 patients with RMC were treated with bevacizumab plus erlotinib, and baseline characteristics are listed in Table 1. The median age at diagnosis of RMC was 31.5 years (range 20–38). Nine patients (90%) were male, seven patients (70%) developed primary disease in the right kidney, and eight patients (80%) previously underwent cytoreductive nephrectomy. At the time of treatment with bevacizumab plus erlotinib, all patients had metastatic disease in the lungs and lymph nodes. Nine patients (90%) had prior platinum-based chemotherapy and the median number of prior therapies was 2.5 (range 0–5). Additional treatment details are shown in Table 2. OncoMine NGS was performed in RMC tumors from 6/10 patients (60%). No somatic alterations were found in five out of six patients (83.3%) whereas one patient (patient BE6) harbored a truncating c.958C > T p.Q320* single nucleotide variation in exon 10 of the NF2 gene.

### 3.2. Efficacy Outcomes

All ten patients had evaluable disease, and Figure 1 shows the waterfall plot of the best-measured response compared with baseline imaging. Two patients (20%) had confirmed partial response (PR) and seven patients (70%) had stable disease (SD) as best overall response per RECIST v1.1. One patient (10%) clinically progressed and died as best overall response. Seven patients (70%) experienced a decrease in tumor burden, including three out of five patients (60%) that had previously received three or more prior lines of therapy. The swimmer plot displaying the types and duration of objective responses to bevacizumab plus erlotinib therapy is shown in Figure 2. At the time of analysis, nine patients (90%) had experienced PD or death, and the median PFS was 3.5 months (95% CI, 1.8–5.2) (Figure 3A). Eight patients (80%) had died, and the median OS1 (overall survival from bevacizumab plus erlotinib initiation) was 7.3 months (95% CI, 0.73–13.8) (Figure 3B). The median OS2 (overall survival from RMC diagnosis) was 20.8 months (95% CI, 14.7–26.8) (Appendix A). The one patient (BE6) harboring a truncating NF2 mutation achieved SD as best response with a PFS of 4.2 months and an OS1 of 22.3 months following bevacizumab plus erlotinib initiation. One patient (BE10) who achieved a PR with fourth-line bevacizumab 10 mg/kg every two weeks plus erlotinib 150 mg by mouth (PO) daily is maintained on this regimen to date, with continued response 10 months from treatment initiation.

### 3.3. Safety Outcomes

Six patients (60%) began therapy with bevacizumab 10 mg/kg intravenously (IV) every 2 weeks in combination with erlotinib 150 mg PO daily. Four patients (40%) began with bevacizumab 15 mg/kg IV every 3 weeks in combination with erlotinib 150 mg PO daily. Treatment-related adverse events are summarized in Table 3. One (10%) grade 3 skin rash required a dose reduction of erlotinib to 100 mg PO daily. No grade 4 or 5 adverse events were attributed to bevacizumab plus erlotinib. The most frequent grade 1–2 adverse events of interest were skin rash (80%), fatigue (30%), hypertension (10%), and proteinuria (10%).

## 4. Discussion

The present study provides, for the first time, evidence that bevacizumab plus erlotinib is clinically active and well tolerated even in heavily pre-treated pts with RMC. The majority of patients had progressive disease after two or more prior lines of treatment, including platinum-based chemotherapy, which is currently the standard of care therapy for RMC but produces responses in only 29% of patients [2,3]. Bevacizumab plus erlotinib produced an objective response rate of 20%, a decrease in tumor burden in 70% of patients, and a disease control rate of 90%. The median PFS benefit of 3.5 months and overall survival (OS) of 7.2 months following bevacizumab plus erlotinib should be interpreted within the context of how aggressive RMC is. The median OS achieved with cytotoxic chemotherapy regimens was only 13 months from RMC diagnosis in a retrospective analysis led by our institution [2]. In the present cohort, the median OS from RMC diagnosis was 20.8 months, suggesting that the addition of bevacizumab plus erlotinib to our therapeutic armamentarium (with 7/10 patients receiving this regimen as their final therapy after progressing on all other options) may prolong survival compared to historical controls. Bevacizumab plus erlotinib was also well tolerated, as only one patient (10%) required a dose reduction, and there were no grade 4 or 5 treatment-related adverse events.

The activity of bevacizumab plus erlotinib has been previously demonstrated in patients with HLRCC and sporadic papillary renal cell carcinoma (spRCC), with particularly potent activity noted in the HLRCC cohort [8,10,16]. HLRCC is an autosomal dominant condition characterized by germline mutations in the fumarate hydratase (FH) gene, and susceptible individuals are at risk for developing cutaneous and uterine leiomyomas, as well as an aggressive form of type 2 papillary renal cell carcinoma [17,18]. Loss of FH alters the TCA cycle and leads to a reliance on aerobic glycolysis and the accumulation of fumarate. Excess fumarate inhibits hypoxia-inducible factor (HIF) prolyl hydroxylase, leading to increased levels of HIF and the subsequent transcriptional upregulation of multiple oncogenic genes, including VEGF and EGFR [13]. Results from a Phase II study (NCT01130519) of bevacizumab plus erlotinib in patients with HLRCC and spRCC demonstrated that this combination is an effective strategy to exploit these unique metabolic vulnerabilities [10]. More specifically, bevacizumab plus erlotinib produced an overall response rate (ORR) of 72.1% (95% CI, 57.2–83.4) in 43 patients with HLRCC and an ORR of 35% (95% CI, 22.1–50.6) in 40 patients with spRCC [10]. The median PFS was 21.1 months (95% CI, 15.6–26.6) in patients with HLRCC and 8.8 months (95% CI, 5.5–15.6) in patients with spRCC [10].

Given the rarity of RMC, there are few published data on the efficacy of treatment regimens for this disease [3]. Furthermore, therapeutic options after disease progression on platinum-based chemotherapy are limited, and novel treatment strategies are desperately needed. The activity of bevacizumab plus erlotinib in our heavily pre-treated cohort substantiates this combination as a new and viable treatment option for patients with RMC. It should be noted that platinum-based chemotherapy can produce durable complete responses in a small subset of patients with RMC [2,3], whereas no such signal was detected in the present study for bevacizumab plus erlotinib. Furthermore, the excellent tolerability we noted with bevacizumab plus erlotinib supports its use in patients burdened by the sequela of prior chemotherapy treatments. We accordingly recommend the use of bevacizumab 10 mg/kg IV every 2 weeks plus erlotinib 150 mg PO daily as a salvage strategy for patients with RMC that develop progressive disease following cytotoxic chemotherapy regimens, most commonly consisting of first-line platinum-based cytotoxic combinations followed by second- or third-line non-platinum chemotherapy agents.

The lack of detectable somatic alterations by clinical-grade NGS assays is typical for RMC [7,19]. This precludes the ability of clinicians to identify molecular subgroups predictive of sensitivity or resistance to bevacizumab plus erlotinib. Further research will be necessary to determine whether the acquired resistance that RMC tumors develop to bevacizumab plus erlotinib is achieved via genomic mutations or other biological processes. Although our study has provided evidence suggesting the clinical value of targeting glucose uptake and utilization in RMC, it remains to be determined whether different approaches can produce more potent and durable responses by more efficiently targeting these metabolic pathways. It should also be noted that our study was not designed to determine the relative efficacy of each of the drugs used. For example, given the substantial upregulation of EGFR pathway signaling in RMC tissues (Appendix A), it is possible that erlotinib produced most of the antitumor effects we observed in our patients. Future preclinical and clinical studies should investigate these questions.

This study has limitations as a single-center retrospective study at a high-volume academic medical center in the United States. Our efficacy findings are prone to selection bias and our safety findings are limited by clinician documentation. The strengths of our study include independent radiographic review to determine ORR and PFS, and a relatively large sample size for this very rare disease.

## 5. Conclusions

In this retrospective study, bevacizumab plus erlotinib showed promising clinical activity and was well tolerated in heavily pre-treated patients with RMC. Although RMC has previously been considered refractory to targeted therapies, bevacizumab plus erlotinib can be used as a salvage regimen to induce responses, improve PFS, and prolong survival compared to previous benchmarks for this deadly disease. However, patients ultimately relapse, and further research is needed to elucidate mechanisms of resistance and to determine how to optimally target metabolic vulnerabilities in RMC.

## Figures and Tables

**Figure 1 cancers-13-02170-f001:**
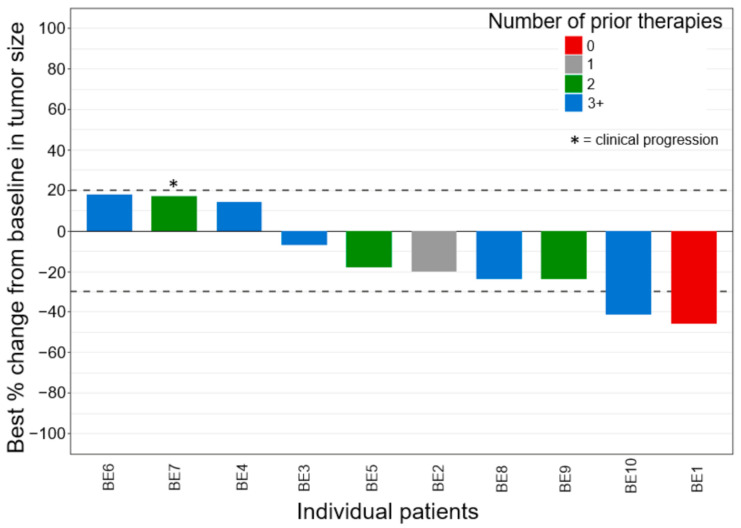
Waterfall plot of best overall response, sorted by number of therapies before bevacizumab + erlotinib. Two patients achieved PR, seven patients achieved SD, and one patient (*) did not meet PD per RECIST v1.1 but clinically progressed and died. PR, partial response; SD, stable disease; PD, progressive disease.

**Figure 2 cancers-13-02170-f002:**
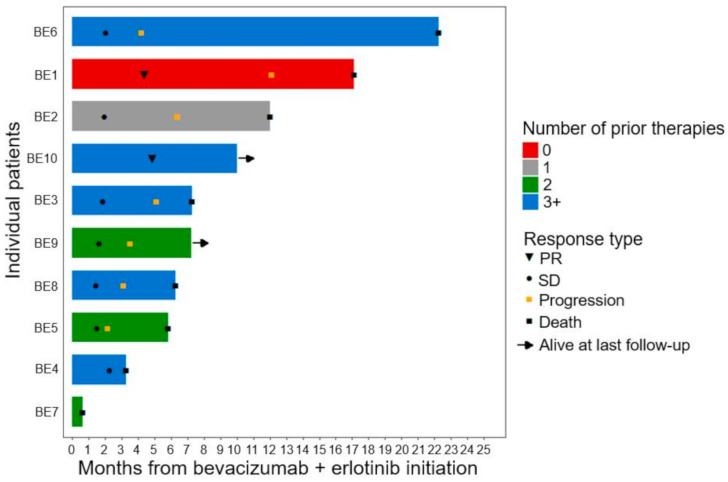
Duration of bevacizumab plus erlotinib. All 10 patients were included in this swimmer plot for duration of therapy, sorted by number of therapies before bevacizumab plus erlotinib. PR, partial response; SD, stable disease.

**Figure 3 cancers-13-02170-f003:**
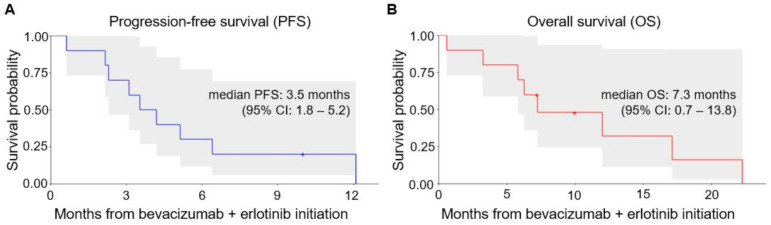
(**A**) Progression-free survival and (**B**) overall survival from treatment initiation with bevacizumab plus erlotinib. The shaded areas represent the 95% confidence bands for each curve. CI, Confidence interval.

**Table 1 cancers-13-02170-t001:** Baseline demographics, clinical and prior treatment characteristics.

Characteristic	Total (n = 10)
Age (Years) at B + E Initiation	
Median	31.5
IQR	23–36
Range (minimum–maximum)	20–37
Gender—no. (%)	
Male	9 (90%)
Female	1 (10%)
Race	
Black	9 (90%)
White, non-Hispanic	1 (10%)
Sickle hemoglobinopathy	
Sickle cell trait	10 (100%)
ECOG performance status at B + E initiation—no. (%)	
0	1 (10%)
1	8 (80%)
2	1 (10%)
RMC laterality—no. (%)	
Left kidney	3 (30%)
Right kidney	7 (70%)
Stage at initial diagnosis of RMC—no. (%)	
I–III	0 (0%)
IV	10 (100%)
Prior cytoreductive nephrectomy—no. (%)	
Yes	8 (80%)
No	2 (20%)
Sites of metastatic disease at B + E initiation—no (%)	
Lung	10 (100%)
Lymph node(s)	10 (100%)
Bone	4 (40%)
Liver	3 (30%)
Other	8 (80%)
Three or more sites of metastatic disease—no. (%)	9 (90%)
Local therapy for metastases—no. (%)	
Metastasectomy	0 (0%)
Radiation	2 (20%)
Prior platinum-based chemotherapy—no. (%)	
Yes	9 (90%)
No	1 (10%)
Number of prior systemic treatments	
Median	2.5
IQR	2–4
Range (minimum–maximum)	0–5
Next generation sequencing of RMC tissues—no. (%)	
Yes	6 (60%)
No	4 (40%)

B + E, bevacizumab plus erlotinib; ECOG, Eastern Cooperative Oncology Group; RMC, renal medullary carcinoma.

**Table 2 cancers-13-02170-t002:** Treatment regimen summary and tumor mutation analysis results.

Patient ID	Treatment #1	Treatment #2	Treatment #3	Treatment #4	Treatment #5	Treatment #6	Treatment #7	NGS
BE1	B + E	Sunitinib	CGB	-	-	-	-	N/A
BE2	TC	B + E	CGB	-	-	-	-	N/A
BE3	GD	GC + Imatinib	ddMVAC	GD + Ifos	B + E	-	-	N/A
BE4	GDI	Ipi/Nivo	TC	CGI	B + E	-	-	None
BE5	TC	GDI	B + E	-	-	-	-	None
BE6	T	Pembro	TC	B + E	VVC	GDI	CGI	NF2
BE7	TC	GDI	B + E	-	-	-	-	None
BE8	TC	T	GDI	Ipi/Nivo	CGI	B+E	-	None
BE9	TC	GDI	B + E	---	-	-	-	None
BE10	Pembro	TC	GDI	B + E	-	-	-	N/A

NGS, OncoMine next generation sequencing; N/A, not available; B + E, bevacizumab plus erlotinib; CGB, capecitabine, gemcitabine, and bevacizumab; TC, paclitaxel and carboplatin; GD, gemcitabine and doxorubicin; GC, gemcitabine and cisplatin; ddMVAC, dose dense methotrexate, vincristine, doxorubicin, and cisplatin; Ifos, ifosfamide; GDI, gemcitabine, doxorubicin, and ixazomib; Ipi/Nivo, ipilimumab plus nivolumab; CGI, cisplatin, gemcitabine, and ifosfamide; T, tazemetostat; Pembro, pembrolizumab; VVC, veliparib, Vx-970, and cisplatin; NF2, truncating NF2 single nucleotide variation.

**Table 3 cancers-13-02170-t003:** Treatment-related adverse events.

CTCAE Term	All Grades	Grade 3
All events	14	1 (7.1%)
Acneiform rash	8	1 (12.5%)
Anorexia	1	0
Fatigue	3	0
Hypertension	1	0
Proteinuria	1	0

CTCAE, Common Terminology Criteria for Adverse Events.

## Data Availability

The data presented in this study are available on request from the corresponding author.

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
