# Peer review of "Efficacy and Safety of Bevacizumab Plus Erlotinib in Patients with Renal Medullary Carcinoma"

_cancers, 2021, doi:10.3390/cancers13092170_

Round 1

Reviewer 1 Report

Summary: Case Series of 10 patients with RMC who all receive bevacizumab and erlotinib – varied from 1st line to 6th line.

I very much enjoyed reading this article as any time I can learn something new about this very lethal disease it helps move patient care forward.

Strengths:

  1. Very little literature on the management of RMC. Therefore any therapeutic guidance is helpful and 10 pts would be considered a "large" series for RMC. 
  2. Blinded review of imaging and thus independent assessment of responses is a big strength.
  3. Paper is reported in a very clear and concise way.  
  4. Figure one contains alot of information and indicating the number of prior therapies in Figure 1 is very helpful.
  5. sound discussion with regards to recommending this as a second line therapy and not first line given the ocassional very good response to platin based therapy. 

Suggestions:

  1. It may be worthwhile commenting on patient BE10 who had a PR, no PD, and is still alive. Are they still on treatment for eg, median f/u of that patient?
  2. Discussion: Line 194-197…”the addition of bev and erolitinib…increased the median OS…to 20.8 m…” might be an overstatement. That is obviously the hope, but there may have been other factors involved as well for eg. these are 10 pts from one institute and thus a more recent cohort than your 2017 multi-institutional publication that you are comparing to.
  3. Conclusion: line 252-253 – as per point 2 above..may want to say something like “improved PFS and prolonged survival compared to historical controls or previous benchmarks in this deadly disease.”

thank you.

Author Response

Reviewer comment: 1. It may be worthwhile commenting on patient BE10 who had a PR, no PD, and is still alive. Are they still on treatment for eg, median f/u of that patient?

Response: Patient BE10 initiated fourth-line therapy with bevacizumab 10mg/kg 2 weeks plus erlotinib 150mg daily and remains on this dose with continued partial response and a follow-up of 10 months. This information has now been added to the results section. 

Reviewer comment: 2. Discussion: Line 194-197…”the addition of bev and erolitinib…increased the median OS…to 20.8 m…” might be an overstatement. That is obviously the hope, but there may have been other factors involved as well for eg. these are 10 pts from one institute and thus a more recent cohort than your 2017 multi-institutional publication that you are comparing to.

Response: Thank you for the valuable feedback. The wording has accordingly been changed to: “In the present cohort, the median OS from RMC diagnosis was 20.8 months, suggesting that the addition of bevacizumab plus erlotinib to our therapeutic armamentarium (with 7/10 patients receiving this regimen as their final therapy after progressing on all other options) may prolong survival compared to historical controls.”

Reviewer comment: 3. Conclusion: line 252-253 – as per point 2 above..may want to say something like “improved PFS and prolonged survival compared to historical controls or previous benchmarks in this deadly disease.”

Response: We appreciate the suggestion. The wording has accordingly been changed to: “Although RMC has previously been considered refractory to targeted therapies, bevacizumab plus erlotinib can be used as a salvage regimen to induce responses, improve PFS, and prolong survival compared to previous benchmarks for this deadly disease.”

Reviewer 2 Report

The authors retrospectively describe 10 patients with pre-treated metastatic renal medullary cancer treated with bevacizumab and erlotinib over the course of 15 years. They found 2 patients with partial response and 7 patients with stable disease. Three patients had progression-free survival exceeding 6 months. Tolerance was very good with only 1 account of grade 3 toxicity (rash).

Although it consists of retrospective data and very small patient numbers, this is normal for such a rare tumor (<0.5% of all renal cancers). Since this tumor is encountered in young patients (20-35 years old), finding treatment options that extend survival are of utmost importance.

Therefore, the current manuscript adds to the (very scarce) literature on this subject.

The manuscript provides well-presented results and is very well written.

I only have set of questions regarding table 2:

All patients had at least 3 different treatment regimens, but these regimens differ in almost all cases. Carboplatinum+paclitaxel, followed by gemcitabine+doxorubicin+ixazomib and then by bevacizumab+erlotinib seems to be the combination used most often (3/10 patients). I suppose RMC being such a rare disease and a time frame spanning over 15 years being the reasons why there was no clear-cut regimen for these patients.

*What were the reasons for the different treatment regiments for all these patients ?

*At this moment, what are the criteria used to select the next treatment regimen for each patient ?

*What (if any) is your current (proposal for) sequence of treatment regimes for a patient presenting with M1 RMC ? This would also be a relevant addition to the manuscript’s discussion.

Author Response

Reviewer: What were the reasons for the different treatment regiments for all these patients ?

Response: Thank you for noting this and it is indeed correct that the treatment heterogeneity reflects the long time frame of the study. 

Reviewer: At this moment, what are the criteria used to select the next treatment regimen for each patient ?

Response: This is an excellent question. Platinum-based cytotoxic chemotherapy has the highest number of durable complete responses (5-10% of cases) in the first-line setting for RMC, and remains the standard of care first-line treatment choice. For platinum-refractory patients, other cytotoxic chemotherapy regimens can be considered as used in Shah et al. BJU Int 2017;120:782-92. This may include novel chemotherapy combinations as part of clinical trials such as our ongoing trial testing gemcitabine plus doxorubicin and ixazomib in patients with RMC (NCT03587662 at clinicaltrials.gov). Our results shown in the present manuscript suggest that bevacizumab plus erlotinib can produce responses even in patients refractory to multiple chemotherapy regimens. In addition, bevacizumab plus erlotinib is a very well tolerated regimen that can be used even in patients with lower performance status following multiple therapies. It is for these reasons that bevacizumab plus erlotinib is currently our preferred third- or fourth-line regimen for patients with RMC. These considerations have now been clarified in the discussion section. 

Reviewer: What (if any) is your current (proposal for) sequence of treatment regimes for a patient presenting with M1 RMC ? This would also be a relevant addition to the manuscript’s discussion.

Response: Thank you for the valuable suggestion. We have accordingly added that our preferred treatment sequence is currently to start with platinum-based chemotherapy, followed by non-platinum chemotherapy regimens, followed by bevacizumab plus erlotinib.